PRDAGE: a prescription recommendation framework for traditional Chinese medicine based on data augmentation and multi-graph embedding

Wen Zhihua 1
Dong Yunchun 2
Peng Lihong 1
Zhang Longxin 1
Yan Junfeng junfengyan@hnucm.edu.cn 2
1 School of Computer Science, Hunan University of Technology , Zhuzhou , China
2 School of Informatics, Hunan University of Chinese Medicine , Changsha , China
Alatas Bilal
Electronic publication date: 2025 Aug 6
Publication date: 2025
Volume: 11
Electronic Location ID: e2974
Received 2024 Nov 27; Accepted 2025 May 29
Copyright: ©2025 Wen et al.
Copyright year: 2025
Copyright holder: Wen et al.
License: This is an open access article distributed under the terms of the Creative Commons Attribution License, which permits unrestricted use, distribution, reproduction and adaptation in any medium and for any purpose provided that it is properly attributed. For attribution, the original author(s), title, publication source (PeerJ Computer Science) and either DOI or URL of the article must be cited.
License URL: https://creativecommons.org/licenses/by/4.0/

Keywords: Prescription recommendation, Herb recommendation, Data augmentation, Symptom-herb relationship representation

Funding: The Key Scientific Research Program of Hunan Provincial Education Department No. 23A0312 The General Scientific Research Program of Hunan Provincial Education Department No. 23C0181 The Natural Science Fundation of Hunan Province No. 2024JJ7143 This research was funded by the Key Scientific Research Program of Hunan Provincial Education Department (No. 23A0312), the General Scientific Research Program of Hunan Provincial Education Department (No. 23C0181), and the Natural Science Fundation of Hunan Province (No. 2024JJ7143). The funders had no role in study design, data collection and analysis, decision to publish, or preparation of the manuscript.

==============================
Background

The prescriptions of traditional chinese medicine (TCM) have made a great contribution to the treatment of disease and the maintenance of good health. Current research on prescription recommendations mainly focuses on the correlation between symptoms and herbs. However, the semantic information inherent in both symptoms and herbs has received limited attention. Furthermore, most datasets in the field of TCM suffer from limited data volumes, which can adversely impact model training.

Methods

To tackle these challenges, we present a prescription recommendation framework called PRDAGE, which is based on data augmentation and multi-graph embedding. We started by collecting medical records and creating a dataset of 3,052 classic medical cases, where we normalized the symptoms and herbs. Additionally, we developed a multi-layer embedding method for symptoms and herbs, using Sentence Bert (SBert) and graph convolutional networks. The aim of this multi-layer embedding method is to capture and represent the semantic information of symptoms and herbs, as well as the complex relationships between them. Additionally, a median-based random data augmentation method was introduced to enrich the medical case data, effectively enhancing the model’s accuracy.

Results

The model was evaluated against baseline models on an unenhanced dataset (Dataset-B), and the results showed that the proposed PRDAGE framework exhibited superior overall performance. Compared to the second-best model, PRDAGE achieved improvements in accuracy and recall rates of 1.69% and 3.80%, respectively, on the Top@10 metric. Ablation experiments further revealed that both the data augmentation and multi-layer embedding modules contributed to the improved model performance.

Conclusion

In conclusion, the experimental results suggest that PRDAGE is an effective prescription recommendation framework. The multi-layer embedding approach effectively represents the semantic information of symptoms and the complex relationships between symptoms and herbs. Additionally, the use of median-based data augmentation has a positive impact on the overall performance and generalization ability of the model.

Introduction

At present, approximately 80% of the global population uses herbal medicine directly or indirectly to control or treat diseases (Izah et al., 2024). Traditional Chinese medicine (TCM), which encompasses a variety of therapeutic approaches including acupuncture and herbal medicine, is an important component of complementary and alternative medicine. It has accumulated extensive empirical knowledge and a vast amount of data in the process of preventing and treating various diseases and is gaining increasing popularity worldwide (Li, Wang & Chen, 2024; Uzuner et al., 2010). Doctors gather information about patient symptoms through observation, auscultation, olfaction, inquiry, and palpation before prescribing a treatment plan. However, the therapeutic process is time-consuming, cumbersome, and heavily reliant on individual doctors’ experience. This poses significant challenges that hinder the further development and popularization of TCM.

With the rapid evolution of deep learning technologies, the application of artificial intelligence in TCM research has emerged as a hotspot in academic circles. Researchers aim to assist doctors in clinical diagnosis and improve treatment efficiency by leveraging artificial intelligence (AI) techniques to automatically recommend Chinese herbal prescriptions based on patients’ symptom descriptions (Zhou, Liu & Wu, 2005; Zhou et al., 2021). Earlier attempts involved training models directly using medical case texts to recommend herbal prescriptions (Zhang et al., 2011; Mi et al., 2015), but the results were not significant. In TCM, doctors diagnose patients based on their symptoms, determine the syndrome type, propose a treatment approach, and prescribe the appropriate herbs. The process can be simplified as a task of solving for herbal prescriptions based on symptom information, once the intermediate black-box thinking steps are overlooked. Researchers have conducted in-depth studies on the process of tasks in recommendation systems, commonly known as prescription recommendation or herbal recommendation (Wang et al., 2019; Dong et al., 2021; Jin et al., 2020).

In the task of prescription recommendation, the quality of symptom embeddings is crucial for effective model training. Extensive research has been conducted in this area (Jin et al., 2021; Jin et al., 2023; Yang et al., 2022) to obtain optimal embeddings for symptoms and herbs. However, this remains a long-standing challenge in prescription recommendation. Representing and capturing the semantic information in symptom texts, the four qi and five flavors of herbs defined in traditional Chinese medicine (Chen, Chen & Crampton, 2004; World Health Organization, 2022), and the complex relationships between symptoms and herbs can be challenging.

Additionally, acquiring high-quality medical case data is a major obstacle in prescription recommendation. The data used to train models typically originates from clinical records of highly skilled TCM experts or is extracted from publicly published medical cases of renowned TCM practitioners. However, the number of carefully curated medical cases by TCM experts is relatively limited, and there is a significant imbalance in the distribution of medical case samples. A significant portion of sample types have a limited number of instances, which does not meet the requirements for training deep learning models. To address this issue, this paper draws inspiration from the concept of text data augmentation. It treats input symptoms as sentences and alters the word order to create new sentences, thereby achieving data augmentation.

In the field of TCM, specific challenges such as the complexity of symptoms and the imbalance of medical case data have hindered the application of deep learning. Traditional manual methods often rely on the experience of TCM practitioners, which is subjective and time-consuming. Previous AI attempts have also faced limitations due to the lack of high-quality TCM data and the complexity of TCM theory. Our study aims to address these issues by proposing the PRDAGE framework that leverages multi-level embedding and data augmentation to improve the performance of TCM prescription recommendation. The paper’s main contributions are:

• A dataset of classical medical cases was extracted and constructed from renowned TCM practitioners. A method for medical case data augmentation was proposed and the optimal coefficient for data augmentation was experimentally validated.

• A multi-level embedding approach for symptoms and herbs was introduced. It effectively captures the semantic information of symptom texts and represents the four qi and five flavors of herbs.

• The PRDAGE framework for recommending Chinese herbal prescriptions is presented. Experimental results on a dataset of classical medical cases demonstrate that PRDAGE outperforms baseline models in terms of recommendation effectiveness.

Related Work

Prescription recommendation tasks are typically implemented through multi-label classification, translation model-based generation, and other methods such as topic modeling and clustering. Clustering is a commonly used technique in prescription recommendation. Chen et al. (2019) and  Qin & Ma (2020) employed clustering algorithms to obtain TCM knowledge and generate prescription recommendations based on symptoms. Topic modelling is a commonly used method for studying TCM prescription recommendations (Lin, Xiahou & Xu, 2016; Ji et al., 2017; Wood et al., 2017). This is achieved by calculating the distribution of symptoms and herbs to discover potential topics. To improve prescription recommendations based on TCM theory, Yao et al. (2018) incorporated domain knowledge into the topic model, proposing a model that represents the TCM theory prescription generation process. However, short text of symptom information tends to perform poorly in topic models. Researchers have conducted end-to-end research on TCM herb generation (Liu et al., 2019; Li et al., 2020; Rong et al., 2022), inspired by natural language tasks such as machine translation. Li & Yang (2019) investigated an end-to-end solution for generating TCM prescriptions using the seq2seq model. Hou et al. (2023) proposed a dual-branch guidance strategy combined with the candidate attention model (DGSCAM) to generate TCM prescriptions automatically based on symptom text. Zhao et al. (2023) introduced a novel hierarchical retrieval mechanism and constructed an intelligent TCM prescription generation model (PreGenerator). However, end-to-end generation models often overlook the complex semantic relationships between symptoms and herbs, as well as their inherent meanings.

Therefore, many scholars adopt multi-label classification to handle prescription recommendation tasks. For instance, Yang et al. (2018) proposed a multi-stage analysis method that integrates tendency case matching, complex network analysis, and TCM compound analysis. This method can identify specific diseases and provide effective TCM prescription recommendations. However, accurate symptom extraction remains a challenging task. Dong et al. (2021) proposed a subnetwork-based symptom term mapping method (SSTM) to represent clinical symptom terminology features and constructed a TCM prescription recommendation framework (TCMPR) using this method. The classification performance of the model is significantly impacted by the embedding of symptoms and herbs, and many scholars have conducted in-depth research on this topic. Jin et al. (2020), Jin et al. (2021) and Jin et al. (2023) constructed symptom-herb heterogeneous graphs and symptom-symptom, herb-herb homogeneous graphs, and used graph convolution, graph attention networks, TCM knowledge graphs, and meta-paths to capture symptom and herb embedding features, achieving good results. In order to capture more complex relationships between symptoms and herbs, researchers (Yang et al., 2022) constructed a TCM knowledge graph. They introduced herb attributes as additional auxiliary information and proposed a multi-layer information fusion graph convolution model. This resulted in information-rich, low-noise symptom feature representation and herb feature representation. Researchers have proposed various models for TCM herb recommendation. For instance, Yang & Ding (2023) proposed a multi-graph residual attention network and semantic knowledge fusion (SMRGAT) model, which uses multi-head attention mechanisms to focus on the different effects of herbs on symptoms. Zhao et al. (2022) proposed a multi-graph convolutional network (MGCN) prescription recommendation model, which considers multiple relationships such as syndrome, syndrome elements, syndrome types, and herbs. Traditional Chinese medicine graph convolutional network (TCM-GCN) (Junfeng, Zhihua & Beiji, 2022) is a method for representing symptoms and herbal medicines that captures their complex relationships. It was previously proposed by us and has yielded promising results in representing these elements in the classic text Treatise on Cold Damage Diseases.

The current mainstream approach for prescription recommendation involves constructing multi-graphs using symptom and herb information from medical records, learning symptom and herb embeddings from the graphs, and then inputting the embeddings into downstream recommendation processes. This approach allows for the capture of co-occurrence information of symptoms and herbs, as well as their associations. However, learning the semantic information from symptom and herb texts through the graphs is not easy. Additionally, the imbalanced sample of medical records in the dataset makes it difficult for rare categories to obtain sufficient training, which severely affects the prediction performance. Thus, this paper proposes the DAGEPR framework, whose main feature is the integration of multi-level symptom and herb embedding methods, followed by the use of TCM-GCN to capture the complex relationships between symptoms and herbs. Specifically, at the symptom level, the text of symptoms is embedded based on a pre-trained model. At the herb level, herbs are embedded based on their attribute information. In the multi-graph layer, the embedded information from the former is fed into TCM-GCN to further capture complex relationships and form new symptom and herb embeddings. Additionally, a random augmentation method based on the median coefficient is proposed to augment the medical case data, which is used to construct the prescription recommendation model and has achieved good results.

Methods

Problem definition

The dataset comprises of classical TCM prescription records that contain essential information, such as patient symptoms, formulas and herbs. Let D denote the dataset, which consists of L cases, M symptoms, and K herbs. C = {c1, c2, …, cL} represents the set of all cases in D, where L is the size of the case set. S = {s1, s2, …, sM} is the set of all symptoms in the dataset, where M is the size of the symptom set. H = {h1, h2, …, hK} denotes the set of all herbs in the dataset, where K is the size of the herb set. In the dataset D, a single case is represented by c=s1,s2,…,si,h1,h2,…,hk and is composed of subsets of S and H. sc=s1,s2,…,si and hc=h1,h2,…,hk denote the symptom set and herb set respectively for that case. Therefore, c=sc,hc can also represent this case.

The diagnostic process in TCM involves extracting symptoms and physical signs from the patient’s body through various methods, such as observation, auscultation, inquiry, and palpation (the four diagnostic methods). The physician then uses a mental model to analyze the extracted information and prescribe a formula that addresses the condition. This task is similar to a prescription recommendation task. The model takes in symptom information as input and outputs a prescription. The learning function of the model is denoted as y=fhC′|Θ, where C′ represents the set of training cases from D, Θ denotes the trainable parameters of the model, and the predicted outcome of y is a probability distribution of herbs(H) with a size of K.

Framework

The PRDAGE framework, shown in Fig. 1, comprises the following components: (1) Data preprocessing module. This module involves primarily extracting medical cases from TCM paper records, extracting vital information such as symptoms and herbs, and standardizing these elements. (2) The symptom and herb embedding module comprises the SBert embedding for symptoms, the representation of the four qi and five flavors, and meridian entry of herbs, as well as the fusion of semantic relation embeddings for symptoms and herbs based on TCM-GCN. (3) The data augmentation module focuses on enhancing symptom-centered case data, while (4) the prescription recommendation module uses the symptom embeddings obtained from (2) and trains with long short-term memory network (LSTM) and fully connected layers to generate the predictive model.

Figure 1 Overall framework of the article.

Firstly, medical case data is extracted from TCM books, and symptoms and herbs are standardized (B). Then, symptoms are embedded using the SBert, and herbs are embedded by their four qi, five flavors, and meridian entry. The symptoms and herbs are input into the TCM-GCN to obtain new symptom and herb embedding enriched with semantic relationships (D). After augmenting the medical case data (C), the data is fed into the prescription recommendation model for herb recommendation (A).

First, after data preprocessing, standardized symptoms and herb names are obtained. The symptoms are fed into the fine-tuned SBert to generate their embeddings, while the herbs are fed into the “Semantic knowledge Embedding” to obtain the embeddings of the four qi, five flavors, and meridian tropism. Subsequently, the embeddings of symptoms and herbs included in the medical case set are fed into the TCM-GCN module, resulting in a set of symptom and herb embeddings that capture the relationships between symptoms and herbs. Second, after the medical case data is fed into the data augmentation module, a medical case feature dataset is generated. Each medical case in this dataset contains several symptom features and corresponding herb features, which are all extracted from the set of symptom and herb embeddings obtained in the first part. Third, based on this medical case feature dataset, it is fed into LSTM and multiple fully connected layers for training to obtain a prediction model. During testing, when n test medical cases are fed in, n herb feature vectors of size m (where m is the size of the entire herb set) are generated. In the generated herb feature vectors, each value represents the predicted probability of the corresponding herb at that position. We select the top K herbs with the highest probabilities as the predicted results for analysis.

Data preprocessing

Data preprocessing is the process of extracting medical case information from TCM classic prescription books. It involves extracting key information such as symptoms, herbs, and formulas for training purposes. The process includes medical case text extraction, symptom and herb entity extraction, data cleaning, and standardization of symptoms and herbs, as shown in Fig. 1B. The primary data source for this study is TCM classic prescription books, including Experimental Records of Classic Prescriptions ( ), Historical Cases of Treatment on Cold Damage ( ), Selected Case Studies of Famous Physicians on Cold Damage Diseases ( ), Collection of Medical Cases on Cold Damage Diseases ( ), Selected Medical Cases on Cold Damage Diseases ( ), Commentary on One Hundred Families’ Medical Cases of the Golden Chamber Prescriptions ( ), Selected Case Studies of Famous Physicians on the Golden Chamber Prescriptions ( ), and Selected Case Studies of National Medical Masters on Classic Prescriptions ( ). We use a CZUR ET16 device to quickly scan the books into images and integrate them into a PDF format with images. Text is recognized using OCR methods, and a total of 3,092 classic medical case data entries are extracted.

To guarantee the quality of the dataset, we cleaned the data collected in by removing duplicate values and discarding cases containing missing or anomalous data. We used named entity recognition methods to extract information such as symptoms from the medical case text. Standardization of symptoms and herbs is necessary to construct a symptom-herb network. The TCM Clinical Common Symptom Terminology Standards (Revised Edition) ( ), Syndrome Differentiation and Treatment Study ( ), and TCM Symptomology Research ( ) were used to standardise all symptoms, while the Chinese Traditional Medicine Thesaurus ( ) and Chinese Herbal Medicine Dictionary ( ) were used to standardise all herbs. Following standardisation, there were a total of 427 symptoms and 429 herbs. Figures 2A and 2B show the frequency distribution of each symptom and herb in the medical cases.

Figure 2 Distribution of symptom and herb frequencies and lengths in medical cases: (A) symptom frequency, mostly below 100 occurrences, with a few exceeding 200, (B) herb frequency, mostly below 50 occurrences, and a few over 250, (C) symptom and herb length distribution, predominantly 5–15 symptoms per case and a normal distribution centered around five herbs.

Embedding of symptom and herb

Establishing prescription recommendation models requires a crucial foundation of embedding symptoms and herbs. To represent the semantic features of symptom texts, we employed an initial embedding using SBert for symptoms and utilized the four qi, five flavors, meridian entry, for embedding herbs. To capture the complex relationships between symptoms and herbs, we utilized TCM-GCN for embedding both symptom and herb nodes. The embedding process for symptoms and herbs is depicted in Fig. 1D.

Symptom embedding

Previous research on symptom embedding often used one-hot encoding, which overlooked the semantic information in symptom texts. SBert is a pre-trained language model-based method that can learn the complex relationships and semantic information between words in sentences, generating sentence vector representations with rich semantic information. We fine-tuned the SBert model using a large corpus of TCM texts. The fine-tuned model was employed to capture the complex semantic information between symptoms in TCM texts, generating symptom embeddings as 256-dimensional vectors.

Herb embedding

Herbs include three-dimensional attributes: four qi, five flavors, and meridian entry (Chen, Chen & Crampton, 2004; World Health Organization, 2022). The four qi refer to the cold, hot, warm and cool (including neutral) properties of herbs, reflecting their tendencies to influence the body’s cold and heat changes. The five flavors of herbs refer to the five distinct tastes of sour, bitter, sweet, pungent, and salty, which are a high-level abstraction and generalization of the functional roles of the herb. The meridian entry of herbs indicates their specific effects on certain organs and meridians of the body, primarily reflecting the scope of their actions. To better represent this semantic information for herbs, we used the method from Junfeng, Zhihua & Beiji (2022) to embed the herbs, resulting in multi-hot vectors of 23 dimensions.

Embedding with TCM-GCN

To capture the relationships between symptoms and herbs, we constructed three graphs: a symptom-herb heterogeneous relation graph (SH Graph), a symptom-symptom homogeneous graph (SS Graph), and a herb-herb homogeneous graph (HH Graph). We then used graph convolutional networks (GCN) to learn embeddings for both symptom and herb nodes.

(1) Construction of symptom-herb relationship

The SH Graph was created by linking symptoms and herbs present in the same medical case. This was done by connecting symptom node si with herb node hj, as shown in Eq. (1). The SH Graph for the classical formulas was built using this approach by establishing edges from all cases in the dataset. (1) SHgraph=1,ifsi,hjco-occur inc0,otherwise.

Additionally, the SS and HH Graphs were constructed to reveal co-occurrence patterns among symptoms and herb compatibility rules, respectively. The SS graph was constructed by connecting co-occurring symptom nodes si and sj using Eq. (2). (2) SSgraph=1,ifsi,sjco-occur inc0,otherwise.

The HH graph was also constructed using the same method as demonstrated in Eq. (3) for all cases in the dataset. (3) HHgraph=1,ifhi,hjco-occur inc0,otherwise.

(2) Embedding of symptoms and herbs

GCN can learn features from the graph topology and capture the complex relationships between nodes. This makes it suitable for representing the intricate interactions among symptom-herb, symptom-symptom, and herb-herb. In this study, we utilized the TCM-GCN model that we previously proposed to learn the features of symptoms and herbs.

In the learning process of the TCM-GCN model, the message function is defined as: (4) mul=MSGleul−1

where mul denotes the message for node u at the l-th layer of propagation, computed from the (l − 1)-th feature vector eul−1 of node u using the message function MSGl, which is a general linear transformation, as follows: (5) mul=Wleul−1

where eul−1 is the (l − 1)-th order feature vector of node u, and Wl is the weight matrix. The multiplication of the two results in the transformed message mul.

The l-th layer’s aggregated information for node v, denoted by N(v), is defined as: (6) evl=AGGlmul,u∈Nv

where N(v) is the set of neighbor nodes of node v, mul is the l-th order message of node u, and u is a neighbor node of v. The aggregation function AGG employs the summation function sum(⋅) to aggregate the messages. To address the potential loss of node-specific information during message passing, the self-message transformation is defined as: (7) mvl=Wlevl−1

where evl−1 is the (l − 1)-th order feature vector of node v, and Wl is the weight matrix. The product yields the self-transformed message mvl, which is then aggregated with the information from neighboring nodes using concatenation: (8) evl=σCONCATAGGmul,u∈Nv,mvl

where the nonlinear activation function σ(⋅) ueses the ReLU(⋅) function, and CONCAT(⋅) denotes the concatenation operation.

Using Eqs. (4) to (8), we perform message passing and neighbor aggregation calculations on the SH Graph in Fig. 1D. The symptom feature vector eshi and herb feature vector ehsi are learned from the symptom-herb SH Graph. The symptom feature vector essi is obtained from the SS graph and the herb feature vector ehhi is obtained from the HH graph. The symptom feature vectors eshi and essi, as well as the herb feature vectors ehsi and ehhi, are then fused pairwise using concatenation. The embeddings of symptoms and herbs are completed by obtaining the symptom feature vector si′ and the herb feature vector hi′.

Data augmentation

Another significant portion of our dataset is the formula. A medical case typically corresponds to one formula (with a small number of samples having two formulas). To analyze the relationship between samples and formula labels, we conducted a comparative analysis of the number of medical cases and the number of formulas. The analysis revealed that most of the formula labels are less than 10 samples, and only nine formula labels have more than 40 samples. The details are shown in Table 1. Labels with fewer than five samples account for one-third of the total label count, including 20 labels with a single sample and 26 labels with two samples. The proportion of labels with low sample counts is relatively large, as showed in Fig. 3A.

During the prediction training, symptoms in a medical case are treated as a sentence and input into the LSTM layer for training. As the input for LSTM is a sequential time series, symptom sequences with different orderings are considered distinct sequences. Therefore, the order of symptoms can be adjusted within a medical case to achieve sample data augmentation. To maintain balance among the expanded samples, we use the median value of 30 in Fig. 3C as the anchor point and expand the samples for labels with fewer than 30 samples. The expansion factor is determined by the formula k = Round(30/n), where n represents the current label’s sample size, and Round() denotes rounding down. If the number of symptoms in a sample is less than or equal to 4, augmentation is performed based on the factorial of the number of symptoms in the medical record. For more than four symptoms, random reordering augmentation is applied according to the value of k, as detailed in Algorithm 1 . Following data augmentation, most medical case samples have over 30 samples (as shown in Fig. 3B), and the center of sample distribution shifts from 2 to 30 (as shown in Fig. 3C). The total number of samples after data augmentation is 9,829.

_______________________ Algorithm 1 Data Augmentation Algorithm____________________________________________ Require: Medical Case Dataset (C = {Ci}|C|     i=1):  A collection of classical pre-      scription medical cases. Ci = {Si,Fi,Hi} is a case in C.Si is a set of symp-      toms present in a medical case, denoted as (s1,s2,...,sm),where si denotes      the ith symptom;  Fi  is the fomula corresponding to this case;  Hi  is the      fomula corresponding to this case.  MAX_NUM, denoting the maximum      anchoring factor, taking values 30, 40, 50, 60. Ensure: The Dataset C′ after augmentation   1:  C′ = ∅  2:  calculate the count of samples for Fi, the result is stored in f_count, which      is a dictionary type variable   3:  for each Ci in C do  4:     if f_count(Fi) <= MAX_NUM then  5:         K = Roundup(MAX_NUM/|Si|)   6:     end if  7:     if K > MAX_NUM//2 and |Si| <= 4 then  8:         Ci′ = itertools.permutations(C i)   9:         C′ = C′∪ Ci′ 10:     else 11:         for i in range(K) do 12:            S′i = random.shuffle((s1,s2,...,sm)) 13:            C′ = C′∪{  Si′,F i,Hi} 14:         end for 15:     end if 16:  end for 17:  return C′_______________________________________________

Prescription recommendations

We input multiple symptoms of a medical case as a sentence into the LSTM layer, and then pass the output to two fully connected layers for training. The output of the fully connected layers, after being processed by the Softmax function, yields the probabilities of the herbs. We select the top K herbs with the highest probabilities as the predicted prescription.

Prescription recommendation is the process of predicting a set of herbs given an input set of symptoms. This task can be understood as calculating the probability of each herb based on a given set of symptoms. It is essentially a multi-label classification task, and therefore we use the binary cross-entropy loss function for model training, as shown in Eq. (9). (9) Ly,y ˆ=−1N ∑i=1Nyi logy ˆi+1−yilog1−y ˆiy ˆ=fhc|Θ

where L represents the loss function, N is the number of medical case samples in the training set, yi is the set of true herbs corresponding to the ith case, y ˆi is the predicted probability of herbs for the ith case, c is the dataset of traditional herbal formulas, and Θ represents the trainable parameters of the model. The predicted result is a set of herb prediction probabilities of size K, denoted as H.

Table 1 Relationship between sample size and number of labels.

Sample size range	≤5	5  < n ≤  10	10  < n ≤  15	15 < n ≤  20	
Number of label types	95	52	29	20	
Sample size range	21 < n ≤  25	26 < n ≤  30	31 < n ≤  40	n > 40	
Number of label types	16	9	17	9	

Figure 3 Distribution of sample and prescription quantities.

(A) The distribution of medical case samples before augmentation, with a considerable number of types having a small number of samples. (B) The distribution of medical case samples after augmentation with an enhancement factor of KAug = 30, where the majority of types have more than 30 samples, and only a few types have less than 10 samples. (C) The distribution of sample numbers before and after data augmentation is compared. Before augmentation, the type with a sample number of 2 has the largest proportion. After augmentation, the type with a sample number greater than 30 has the largest proportion.

Experimental Results and Analysis

Data sets

Through the methods described in Data Preprocessing for data collection and preprocessing, we obtained 3,052 medical case data, which is named Dataset-A. After data augmentation with a 30-fold increase, a total of 9,829 data entries were obtained, named Dataset-A-30. The dataset contains 427 different symptoms and 429 different herbs, and is divided into two subsets for training and testing purposes. The training and testing subsets have proportions of 0.7 and 0.3, respectively. To test the model’s ability to generalize to real medical cases, we selected medical case data from outside the dataset. We extracted 278 traditional herbal formula medical cases from Selected Medical Cases of Liu Duzhou ( ) and Essential Collection of Hu Xishu’s Medical Cases ( ), and named this dataset Dataset-B. We then conducted secondary testing on Dataset-B to further validate the model’s generalization ability. Table 2 presents the statistical information of the experimental datasets.

Table 2 Statistics of the evaluation dataset.

Dataset	Samples	Symptoms	Herbs	
Dataset-A-30	9,829	427	429	
Dataset A-30 training set	6,880	427	429	
Dataset-A-30 test set	2,949	393	333	
Dataset-B	278	348	157	

Experimental setup

The experiments were conducted using TensorFlow-GPU 2.9.0 on an Intel(R) Xeon(R) Silver 4314 CPU @ 2.40 GHz 2.39 GHz (2 processors) with 32 GB of memory. GPU-accelerated training was performed using NVIDIA Tesla T4. The symptom embeddings were set to 256 dimensions, the LSTM layer dimensions were set to 256, and the fully connected layer dimensions were set to 256 and 512 respectively. The output layer dimensions were consistent with the number of herbs, which was 429. The initial learning rate for the model was set to 1e−4, and the Adam optimizer was used to update the parameters with momentum parameters of β1 = 0.9 and β2 = 0.99. The model was trained for 100 epochs with a batch_size of 8. If the loss and hit rate on the test set remained unchanged after 10 epochs, training was terminated early, and the optimal model would be saved.

Baseline model

In this study, we employ the following baseline models to validate the proposed method:

KGETM (Wang et al., 2019): The Knowledge Graph Embedding Enhanced Topic Model is a model used for herb recommendation. It leverages TransE to learn semantic knowledge of symptoms and herbs in TCM knowledge graphs, constructing a graph embedding method for TCM knowledge.

SMGCN (Jin et al., 2020): The Symptom-Multi-Graph Convolutional Network is a model that constructs multiple graphs between symptoms and herbs to obtain embeddings for symptoms and herbs. It also derives representations of syndromes and trains alongside symptoms to recommend herbs.

KDHR (Yang et al., 2022): A graph convolutional network-based herb recommendation model that integrates auxiliary properties of herbs on the basis of a TCM knowledge graph. It obtains symptom feature and herb feature representations through a ‘symptom-herb’ graph.

TCMPR (Dong et al., 2021): A TCM prescription recommendation method based on a sub-network symptom word mapping approach. It extracts sub-network structures between symptoms from the knowledge network, effectively representing the embedding features of symptom terminology.

PRDAGE: Our proposed model is a prescription recommendation based on data augmentation and multi-graph embedding for TCM (PRDAGE). It is designed to capture the textual semantic information of symptoms and represent the complex relationships between symptoms and herbs.

Evaluation metrics

To quantitatively assess the effectiveness of the PRDAGE method, we employ three performance evaluation metrics: precision, recall, and F1-score, to measure the model’s performance. For medical case c in the test set, the evaluation metrics are defined as follows: (10) Precision@K=|TopSetpre,K∩Setlabel|K

(11) Recall@K=|TopSetpre,K∩Setlabel||Setlabel|

(12) F1−score@K=2∗Precision@K∗Recall@KPrecision@K+Recall@K.

In predicting herbs, Top(Setpre, K) represents the K herbs with the highest prediction scores. Setlabel denotes the actual herbs present in hc. Precision at k (Precision@K) indicates the proportion of correctly predicted herbs among the top K predicted herbs. Recall at k (Recall@K) represents the proportion of the top K predicted herbs that are actually present in hc. The F1-score is the weighted average of the first two metrics, providing a more objective representation of the model’s performance. The value of K ranges from 1 to 20.

Table 3 Performance comparison of PRDAGE and other models.

Method	Precision	Recall	F1-score	
	P@5	P@10	P@20	R@5	R@10	R@20	F1@5	F1@10	F1@20	
KGETM	0.3703	0.2671	0.1693	0.3215	0.4634	0.5802	0.3442	0.3389	0.2621	
SMGCN	0.3851	0.2746	0.1759	0.3416	0.4756	0.5979	0.362	0.3481	0.2719	
KDHR	0.4015	0.2812	0.1826	0.3566	0.4855	0.6211	0.3777	0.3561	0.2822	
TCMPR	0.3945	0.2953	0.1935	0.3510	0.5041	0.6488	0.3715	0.3724	0.2981	
PRDAGE (ours)	0.4046	0.3003	0.1929	0.365	0.5233	0.6663	0.3838	0.3816	0.2992	
Improv. by KGETM	9.26%	12.43%	13.94%	13.53%	12.93%	14.84%	11.50%	12.60%	14.15%	
Improv. by SMGCN	5.06%	9.36%	9.66%	6.85%	10.03%	11.44%	6.02%	9.62%	10.04%	
Improv. by KDHR	0.77%	6.79%	5.64%	2.36%	7.79%	7.28%	1.62%	7.16%	6.02%	
Improv. by TCMPR	2.56%	1.69%	–	3.99%	3.81%	2.70%	3.31%	2.47%	0.37%	
Notes.

Bold and underlined values depict the best and second-best results, respectively.

Experimental results

The experiments were conducted with the model trained on Dataset A-30. To avoid overfitting caused by augmented data, model testing was performed on the unaugmented dataset (Dataset-B). We compared the PRDAGE model with four other models: KGETM, SMGCN, KDHR, and TCMPR. The test results shown in Table 3, where bold and underlined values depict the best and second-best results, respectively. The results indicate that the PRDAGE model demonstrates the best overall performance in Top@5, Top@10, and Top@20. Compared to KGETM, PRDAGE achieved a 9.26% increase in precision, a 13.53% increase in recall, and an 11.50% increase in F1 score at Top@5. At Top@10, the accuracy improved by 12.43%, recall by 12.93%, and F1 score by 12.60%. At Top@20, the accuracy, recall, and F1 score increased by 13.94%, 14.84%, and 14.15%, respectively. These improvements are statistically significant. Although DAGEPR’s performance enhancement is not significant compared to the second-best model, there is still some improvement. Specifically, at Top@5, the accuracy and recall increased by 0.77% and 2.36%, respectively. At Top@10, the accuracy and recall improved by 1.69% and 3.81%, respectively.

Finally, we analyzed GPU memory consumption, computational time and model complexity of DAGEPR and baseline methods. GPU memory consumption and computational time is measured by the GPU memory used by the models during training. Model complexity is evaluated by the number of parameters.

Table 4 shows the comparison between DAGEPR and the four baseline methods in terms of computational time, number of parameters, and GPU memory usage: (1) GPU memory usage: the GPU memory usage of DAGEPR was 852 MB, which was much lower than that of TCMPR, and KDHR. (2) Number of parameters: the number of parameters of DAGEPR was 0.9427 M, which was less than that of TCMPR. (3) Computational Time: the computational time of DAGEPR was 197.27 s, much less than KDHR and TCMPR. In summary, DAGEPR was better than most of the baselines in terms of computational efficiency and resource usage, and the number of parameters was moderate.

Table 4 Comparison of DAGEPR and other models based on GPU memory usage, number of parameters, and computational time.

Method	GPU memory	Params	Time	
KGETM	86 MB	0.0183 M	35.832 s	
SMGCN	827 MB	0.6135 M	112.93 s	
KDHR	916 MB	0.8741 M	231.83 s	
TCMPR	1,173 MB	0.6135 M	112.93 s	
DAGEPR	852 MB	0.9427 M	197.27 s	

Ablation experiments

PRDAGE consists of two crucial components: the embedding module for symptoms and herbs, and the data augmentation module. To validate the effectiveness of these two components in model training, we conducted tests under the following four scenarios (as shown in Table 5): (1) PRDAGE without symptom/herb embedding and data augmentation (PRDAGE-NoE-NoA), (2) PRDAGE without symptom/herb embedding but with data augmentation (PRDAGE-NoE), (3) PRDAGE with symptom/herb embedding but without data augmentation (PRDAGE-NoA), and (4) PRDAGE with both symptom/herb embedding and data augmentation. The experimental comparison results are presented in Fig. 4. It can be observed from the figure that the overall performance is the worst when neither augmentation nor embedding is applied (PRDAGE-NoE-NoA), and the best when both augmentation and embedding are utilized (PRDAGE), indicating that data augmentation and embeddings significantly contribute to the improvement of model performance. The performance with only augmentation (PRDAGE-NoE) and only embedding (PRDAGE-NoA) is relatively close, but the performance with only embedding is slightly better than that with only augmentation, which suggests that the embedding module plays a more important role in enhancing model performance.

Table 5 Ablation experiment modules.

Name	Embedding module	Augmentation module	
PRDAGE-NoE-NoA	0	0	
PRDAGE-NoE	0	1	
PRDAGE-NoA	1	0	
PRDAGE	1	1	

Figure 4 Precision (A), recall (B), and F1 score (C) for the four different scenarios: with both embedding and augmentation (PRDAGE), which performs the best; with embedding but without augmentation (PRDAGE-NoE), which performs second best; without embedding but with augmentation (PRDAGE-NoA), which performs third; and without embedding and without augmentation (PRDAGE-NoE-NoA), which performs the worst.

Performance comparison with different data augmentation factors

To seek the optimal data augmentation coefficient, we tested datasets augmented with different coefficients. We augmented the datasets with various augmentation coefficients, KAug = 10, 20, 30, 40, 50, 60. In order to enhance the training speed, the learning rate was set to 0.0001, and the epoch number was set to 100. The final results are shown in Fig. 5A. It can be observed from the figure that as the augmentation coefficient increases, the model’s accuracy also improves. When KAug = 30, the model achieves optimal performance in terms of precision, recall, and F1 score. However, when the augmentation coefficient is further increased, the overall performance of the model declines. Therefore, we select 30 as the anchor value for the medical case data augmentation coefficient.

Figure 5 Precision, recall, and F1 score of the model on Dataset-B and Dataset-A-K-test.

(A) The generalization capability of different models, i.e., the performance of models trained on datasets augmented by a factor of KAug (Dataset-A-K) when tested on an unaugmented dataset (Dataset-B). (B) The performance of these models on their respective test sets (Dataset-A-K-test).

In order to assess the model’s performance on augmented datasets, the data was augmented with different KAug coefficients, and the dataset was split into training and testing sets at a ratio of 0.7:0.3. The model’s performance on the augmented test set is shown in Fig. 5B, while the results on the unaugmented test set are presented in Fig. 5A. It can be observed from the figures that models trained on augmented datasets achieve over 70% Top@1 precision and Top@20 recall. When the KAug coefficient is 50 and 60, the model’s Top@1 precision even exceeds 90%. However, when tested on unaugmented datasets, the accuracy significantly decreases when the KAug coefficient exceeds 30. This phenomenon can be attributed to the augmented dataset’s increased proportion of similar data, which may result in overfitting during model training.

Data augmentation is an effective method to address data scarcity, but it can also lead to overfitting issues. In this study, several aspects were considered to reduce the risk of overfitting. Firstly, experiments were conducted with different augmentation factors (KAug =10, 20, 30, 40, 50, 60), and it was found that the model achieved the best performance in terms of accuracy, recall, and F1-score when KAug =30. Beyond this value, the model’s performance began to decline, indicating the onset of overfitting due to the introduction of too many similar samples. Secondly, an early stopping strategy was implemented during the training process to prevent overfitting. Lastly, to ensure that our model did not overfit the augmented data, its performance was evaluated on an independent, unaugmented test dataset (Dataset-B). The results showed that the model maintained good performance on this dataset, indicating that it had learned robust features rather than just fitting the augmented data.

Analysis of the overall model performance

The results in Fig. 5A indicate that the selection of an appropriate data augmentation coefficient is crucial. Only models trained with a suitable data augmentation coefficient exhibit good generalization capabilities. Otherwise, there is a risk of the model performing exceptionally well during training but poorly on other datasets.

From Fig. 5, it can be observed that when we consider the number of herbs in the prescription for augmentation (i.e., as Top@K increases), the precision shows a declining trend, while the recall rate increases significantly, and the F1-score also improves. The F1 score increases in the Top@K range of [1–5], remains relatively stable in the range of [5–10], and then shows a marked decline after 10. This trend is likely related to the number of herbs in the dataset samples. In this dataset, the majority of medical case samples have a number of herbs within the range of [5–10], hence the F1 score also reaches its optimum and remains stable in this interval. In the reference [6], the mean number of herbs per prescription in their dataset is 11.3, and their model’s F1 score begins to stabilize from Top@12, which supports our observations.

Although the model achieved optimal results compared to the baseline models, its overall performance remains relatively average, with a noticeable gap from the requirements of clinical practice. Generally, the overall performance of prescription recommendation models is relatively low, which may be attributed to the fact that prescriptions consist of combinations of multiple herbs. Some herbs have similar effects and can substitute for each other, a factor that is not considered in the medical case data and evaluation. Additionally, another reason may be that the amount of medical case samples used for training is not large enough to fully reflect the complex associations between symptoms and herbs. Therefore, in future research, it is necessary to continue collecting medical case data and expand the sample size of medical cases.

Conclusion

This study proposes a TCM prescription recommendation framework based on data augmentation and multi-graph embedding (PRDAGE). A multi-layer embedding approach is adopted for symptoms, herbs, and their relationships to capture their semantic information and complex associations. Additionally, the proposed median random medical case data augmentation method effectively expands the medical case data. Evaluations on both augmented and unaugmented test sets demonstrate that our model’s comprehensive performance is optimal among the baseline models. Furthermore, by testing on datasets augmented with different multipliers, we identified the augmentation coefficient values that yield the best generalisation capabilities for the model. This research is expected to lay the foundation for further studies in intelligent TCM clinical diagnosis and treatment.

However, there are still some limitations in this study. Although the relationship between symptoms and herbs has been established through medical cases, the explicit knowledge of TCM plays a significant role in diagnosis, which has not yet been incorporated into this study. In the future, TCM knowledge graphs can be integrated into model training. Additionally, the weights of chief complaints and the relationships among sovereign, minister, assistant, and messenger herbs in prescriptions have not been considered. These factors will be included in future research on prescription recommendations.

Supplemental Information

Supplemental Information 1 The code of the model and data augmentation, and the total dadaset

The raw data used for training and testing in the dataset are medical case records in Chinese text. This study is trained based on Chinese text. To facilitate non-Chinese researchers, we have provided the English names of the attributes in the dataset and mapped them to their corresponding Chinese names.

The .pickle file can be read using Python language editing software, such as PyCharm. The *.pickle file in this program is a serialized file in Python dictionary format, containing key-value data, where the key is the symptom name and the corresponding value is the symptom embedding vector. The symptom name is the normalized symptom name, and the symptom embedding vector is generated by SBert. You can use the pickle.load() method from the pickle module in the program to restore the file as a Python dictionary variable.

Additional Information and Declarations

Competing Interests

Author Contributions

Data Availability

The authors declare there are no competing interests.

Zhihua Wen conceived and designed the experiments, performed the experiments, analyzed the data, performed the computation work, prepared figures and/or tables, authored or reviewed drafts of the article, and approved the final draft.

Yunchun Dong conceived and designed the experiments, performed the experiments, analyzed the data, performed the computation work, prepared figures and/or tables, and approved the final draft.

Lihong Peng analyzed the data, prepared figures and/or tables, and approved the final draft.

Longxin Zhang analyzed the data, prepared figures and/or tables, and approved the final draft.

Junfeng Yan conceived and designed the experiments, analyzed the data, prepared figures and/or tables, authored or reviewed drafts of the article, and approved the final draft.

The following information was supplied regarding data availability:

Code and raw data are available in the Supplemental Files.

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
