# Peer review of "PRDAGE: a prescription recommendation framework for traditional Chinese medicine based on data augmentation and multi-graph embedding"

_PeerJ Computer Science, doi:10.7717/peerj-cs.2974_

## Round 0.1 · original submission · Major Revisions

Please respond to the comments of all 4 reviewers

·

Basic reporting

Authors have designed a Recommendation framework named PRDAGE, which is based on Data Augmentation and multi-Graph Embedding. To tackle the information of TCM which suffers from limited data volumes, which can adversely impact model training for recommending of Chinese herbal prescriptions .

Experimental design

. The experiment design of manuscript is very good and understandable .

Validity of the findings

Since the PRDAGE model has been developed using experimental data , findings can be repeated and the prediction may be very useful for recommendation of Chinese herbal prescriptions .

Additional comments

It is a well-written manuscript. The title of this manuscript is appropriate. The introduction has been well described. Methods experimental as well computational are written very well. I suppose that readers can reproduce methods very easily. The results and figures are well explained. The discussion and conclusion sections explain very well the outcomes of this manuscript.

Reviewer 2 ·

Basic reporting

The authors have proposed a concept "PRDAGE: A prescription recommendation framework
for traditional Chinese medicine based on data augmentation and multi-graph embedding". This method regulates the prescription recommendation with NLP techniques, LLMs (SBERT), and graph convolutional networks. The concept seems interesting, but the results are not significant enough to justify the performance usable in real-time. Still, we appreciate the efforts of the authors to report the findings appropriately and well-structured.

Experimental design

Include the core experiment as a workflow (micromodules) only with each component representing the blocks. The system architecture looks a bit fuzzy to understand the interaction between the modules.

What are the efforts taken or planned to implement the framework in real time? The future of traditional medicine seems promising due to its wide acceptance across different communities, globally.

Ensuring the reproducibility of the findings or stability of the results is another important area to discuss in the manuscript.

Limitations should be addressed for the proposed work.

As mentioned in the contributions, for the data extracted from the TCM practitioners in real cases, did the authors receive ethical approval? If so, mention it in the contribution.

Suggest how the performance can be improved with additional efforts. In clinical practice, the reliability of the system is an essential factor. The proposed work is underperforming concerning the evaluations.

As GCN is used, isn't there any alternative model available to benchmark the model validity?

Validity of the findings

The findings are relevant to the study, but the results are not rational enough to support the use case in real-world settings. But the results could've been improved.

For Figures 4 and 5, use the same scale (0 to 1) to plot the graphs. Different scales shouldn't be used when the comparison is performed between each other.

The sample outcome for any scenario generated by the PRDAGE system can be attached as a figure.

Additional comments

In Fig.3 (c), the word "before" is misspelled.

Don't use @ to represent the gap, use a hyphen instead or leave a space.

·

Basic reporting

FOR THE INTRODUCTION
1. There are generalized statements about TCM's wide usage. However, there is no quantitative support or specific numbers or examples to back this claim. Add case studies demonstrating its significance globally.

2. TCM-specific challenges lacks deep learning grounding. Please include a brief comparison with existing manual or AI failures in TCM context.

3. The introduction part focuses on East Asian settings. Please expand how the proposed framework could generalized to other healthcare systems.

4. Previous models are mentioned without discussing their limitations. Provide one.

FOR THE LITERATURE REVIEW

5. The literature touches data limitations. Please explore solutions such as transfer learning or synthetic data generation.

6. It is not clear how PRDAGE differs from existing graph-embedding methods. Add examples or instances of its uniqueness.

7. Include in the literature classical TCM researches. Explain how AI modeling would improve between TCM and computational approaches.

Experimental design

FOR METHODOLOGY

8. Explain further the SBERT-fine-tuning for symptoms embedding to ensure reproducibility.

9. Explanation of the seamless integration of the framework's architecture could be improved.

10. Why binary cross-entropy was chosen over other loss functions, what is special about it?

11. There is minimal discussion about potential overfitting, especially with the extensive use of augmentation. Explain further.

12. No discussion is provided about the training duration or resource consumption, which is critical for evaluating the model's feasibility.

13. The framework’s modularity and ease of adaptation for non-TCM datasets are not explored, which limits its broader applicability.

Validity of the findings

FOR RESULTS & DISCUSSIONS

14. Consider providing additional statistical test to confirm whether the improvements in accuracy and recalls are significant.

15. Provide information on computational costs or scalability of PRDAGE compared to simpler models.
Weigh its pros and cons.

16. A multi-graph embedding approach conducted, its specific improvement are not properly quantified. Provide one.

Additional comments

Fix clarity and flow of sentences, including grammars.

Reviewer 4 ·

Basic reporting

-The article uses clear, unambiguous, and professional English throughout.

- It provides adequate background and context from relevant literature with appropriate references to prior research.

- The article follows a professional structure, including figures and tables.

- It is self-contained, presenting relevant results that align with the hypotheses.

Experimental design

No comment

Validity of the findings

-The rationale and benefit to literature are clearly stated.
-All underlying data are provided, robust, statistically sound, and controlled.
-Conclusions: Well stated, directly linked to the original research question, and limited to supporting results.

Additional comments

With Prescription Recommendation framework based on data augmentation and multi-graph embedding (PRDAGE), the authors have successfully described a new method for Traditional Medicine prescription. They suggested a clear description of symptom-herb embedding and multi-dimension data augmentation.
However, the authors can be more mindful of the consistency of the whole theory.

1. Line 164, there is a repeat of phrase: “and standardizing”. Line 370, there is a miss typing of punctuation: (5-10]. Please correct those.
2. Line 169: “LSTM layer” and line 126: “TCM-GCN” is lacking full descriptions for abbreviations as their first appearance in the script. Line 206, TCM should be state instead of Traditional Chinese Medicine.
3. In the Related Work part, the authors can include 1 table or a flow chart which classify and summarize the weakness of old methods (end-to-end models) and point out the novelty of this research. This help the readers follow this work easily.
4. From line 189-192, please provide the year, publisher, or reverences (if possible) of the mentioned TCM’s books.
5. Some figures are not cited in the text (e.g. Figure 3a). Each figure should come after being mentioned in the text which makes them easier to track. Please clarify this.
6. Figure 2 and 3 have too small sub-labels, please correct them.
7. Line 241 and 273, please make sure what is Section * and Section 3.3.
8. In Table 1, why the sample size range is 21<n ≤ 15?

Annotated reviews are not available for download in order to protect the identity of reviewers who chose to remain anonymous.

---

## Round 0.2 · accepted · Accept

Dear Authors,

It is evident that one of the preceding reviewers elected not to respond to the invitation to review the revised paper. It has been observed that the preceding two reviewers have accepted the paper. Following a thorough evaluation, it has been determined that the paper has been sufficiently improved and can be approved in its current state. However, please use equations with correct equation number. Many of the equations are part of the related sentences and attention is needed for correct sentence formation prior to the execution of the production stage.

Best wishes,

·

Basic reporting

1. The paper now has enough details after revisions.

Experimental design

2. Methods have been substantially improved.

Validity of the findings

3. Findings and results are now complete.

Additional comments

4. Recommended for publication.

Reviewer 4 ·

Basic reporting

-The article uses clear, unambiguous, and professional English throughout.
- It provides adequate background and context from relevant literature with appropriate references to prior research.
- The article follows a professional structure, including figures and tables.
- It is self-contained, presenting relevant results that align with the hypotheses.

Experimental design

-

Validity of the findings

-The rationale and benefit to the literature are clearly stated.
-All underlying data are provided, robust, statistically sound, and controlled.
-Conclusions: Well stated, directly linked to the original research question, and limited to supporting results.

Additional comments

Prescription Recommendation framework based on data augmentation and multi-graph embedding (PRDAGE), the authors have successfully described a new method
for Traditional Medicine prescription. They suggested a clear description of symptom
embedding and multi-dimensional data augmentation.

The authors have carefully revised the manuscript following the previous reviewer's comments.